

**Characterization of aerosol hygroscopicity, mixing state, and**
**CCN activity at a suburban site in the central North China Plain**
**Yuying Wang[1], Zhanqing Li[1], Yingjie Zhang[2], Wei Du[2,3], Fang Zhang[1] , Haobo Tan[4],**
**Hanbing Xu[5], Xiaoai Jin[1], Xinxin Fan[1], Zipeng Dong[1], Qiuyan Wang[6], Yele Sun[2,3]**
[1]College of Global Change and Earth System Science, Beijing Normal University, Beijing 100875,
China
[2]State Key Laboratory of Atmospheric Boundary Layer Physics and Atmospheric Chemistry,
Institute of Atmospheric Physics, Chinese Academy of Sciences, Beijing 100029, China
[3]College of Earth Sciences, University of Chinese Academy of Sciences, Beijing 100049, China
[4]Key Laboratory of Regional Numerical Weather Prediction, Institute of Tropical and Marine
Meteorology, China Meteorological Administration, Guangzhou 510080, China
[5]Shared Experimental Education Center, Sun Yat-sen University, Guangzhou 510275, China
[6]Collaborative Innovation Center on Forecast and Evaluation of Meteorological Disasters, Nanjing
University of Information Science and Technology, Nanjing, 210044, China
*Correspondence to*: Zhanqing Li (zli@atmos.umd.edu)



**Abstract.** Aerosol hygroscopicity, mixing state and CCN activity were investigated as
a part of the Atmosphere-Aerosol-Boundary Layer-Cloud ($A^2BC$) Interaction Joint
Experiment carried out at Xingtai (XT), a suburban site in the center of the North
China Plain (NCP). In general, the probability density function of the hygroscopicity
parameter ($\kappa$-PDF) for 40–200 nm particles had a unimodal distribution and mean
$\kappa$-PDF patterns for different sizes were similar, suggesting that the particles were
highly aged and internally mixed because of strong photochemical reactions. The $\kappa$
calculated from the hygroscopic growth factor in the daytime and at nighttime showed
that photochemical reactions largely enhanced the aerosol hygroscopicity, and the
effect became weaker as the particle size increased. In addition, the aerosol
hygroscopicity was much larger at XT than at sites in the northern part of the NCP,
illustrating that the hygroscopicity of particles varies due to different emissions and
chemical processes in the NCP.

Measurement results also showed that new particle formation events occurred

frequently at XT, one of the most polluted city in China. The evolution of the
planetary boundary layer played a dominant role in aerosol mass concentration
changes while particle formation and growth had a greater influence on the variation
in aerosol number concentrations. Particle size was the most important factor
influencing the ability of aerosols to activate, especially at higher levels of
supersaturation (SS). The cloud condensation nuclei (CCN) number concentration
($N_{CCN}$) derived from chemical composition was highly correlated with the measured
$N_{CCN}$ ($R^2 \geq 0.85$), but was generally overestimated due to measurement uncertainties.



The effect of chemical composition on $N_{CCN}$ was weaker relative to the particle size.
$N_{CCN}$ sensitivity tests showed that the impact of chemical composition on $N_{CCN}$
became weaker with increasing SS, suggesting that chemical composition played a
less role in $N_{CCN}$ estimations at higher SS levels. A good proxy for the chemical
comical composition ($\kappa = 0.31$) was found, which can simplify the calculation of
$N_{CCN}$ on models.
**1.   Introduction**
Aerosols, defined as the mixture of solid and liquid particles, are ubiquitously
present in the atmosphere because of direct emissions from biogenic and
anthropogenic sources and the secondary transformation from gas precursors. Aerosol
particles play an important role in climate changes through direct and indirect effects
(e.g. Ramanathan et al., 2001; Daniel et al., 2008; Z. Li et al., 2016). However, the
impact of aerosols on climate change is difficult to simulate because of the highly
variable physical and chemical properties of aerosols, and complex aerosol-cloud
interactions (IPCC, 2013; Lebo et al., 2017).
The hygroscopic growth and mixing state of aerosol particles are important for
estimating the direct climate effect of aerosols. This is because the growth and mixing
can change the particle size and optical properties of aerosol particles, directly
influencing the terrestrial radiation budget and degrading the atmospheric visibility
(Covert et al., 1972; Stock et al., 2011; Peng et al., 2016; Z. Li et al., 2017). In
addition, aerosol particles can be activated as cloud condensation nuclei (CCN) under





supersaturation (SS) conditions. The variability in CCN number concentration ($N_{CCN}$)
can modify cloud microphysical properties, thereby causing an indirect radiative
forcing (Twomey, 1974; Albrecht, 1989). Previous studies have addressed three main
aerosol properties influencing the CCN activation, namely, particle size, chemical
composition, and mixing state. However, their relative importance is different in
different environments (e.g. Ervens et al., 2007; Cubison et al., 2008; Deng et al.,
2011; Zhang et al., 2014).
Ambient aerosols are composed of different species, including inorganic ions,
organic components, black carbon (BC), and mineral dust. Inorganics mainly contain
sulfate, nitrate, and ammonium, while organic aerosols (OA) consist of thousands of
chemicals (Jacobson et al., 2000). The hygroscopicity and CCN activity of a single
component can be characterized according to laboratory studies (e.g. Petters and
Kreidenweis, 2007), but the properties of their mixtures are hard to estimate because
of the different chemical species and mixing states of particles in the atmosphere.
Therefore, aerosol hygroscopicity and CCN activity are very different in different
regions. Comprehensive field measurements of aerosol properties in different areas
are necessary to improve models.
China, especially the North China Plain (NCP), has been suffered from severe air
pollution since its rapid industrialization and urbanization in the last couple of
decades, where diverse sources and aging processes make aerosol properties
particularly diverse and complex. As such, the region has drawn much attention in
studying the aerosol mixing state, hygroscopicity, and CCN activity (Deng et al., 2011;





Liu et al., 2011; Zhang et al., 2014; F. Zhang et al., 2016; S.L. Zhang et al., 2016; Wu
et al., 2016; Y. Wang et al., 2017). Liu et al. (2011) and Y. Wang et al. (2017) have
suggested that ambient particles are mostly an external mixture with different
hygroscopicities. Deng et al. (2011) has shown that the aerosol number size
distribution is critical in the prediction of $N_{CCN}$ while Zhang et al. (2014) have
highlighted the importance of chemical composition in determining particle activation
properties. However, all these studies were done using data from the northern part of
the NCP. Few studies have focused on the central region of the NCP. Compared to the
northern part of the NCP, the central part of the NCP is more affected by industrial
emissions where a dense cluster of China's heavy industries exist (Fu et al., 2014).
Measurement of aerosol properties in the central part of the NCP are critically needed
to investigate the impact of air pollution on the environment and climate changes.
Xingtai (XT), a city located in the middle of the NCP, often ranks in the top of
polluted cities in China. Local industrial and domestic sources are the greatest
contributors to severe haze events (Wang et al., 2014). A field experiment called the
Atmosphere-Aerosol-Boundary Layer-Cloud (A$^2$BC) Interaction Joint Experiment
was carried out at a suburban site in Xingtai in the summer of 2016. Differences in
aerosol properties at this site and at sites in the northern part of the NCP were found in
this study.
The paper is organized as follows. Sections 2 and 3 describe the measurement
method and data analysis theory. Section 4 presents and discusses the measurement
results, which includes the data time series, aerosol mixing state, hygroscopicity, CCN





prediction and its sensitivity to chemical composition. A summary and conclusions are
given in section 5.

## 2.    Measurements

### 2.1.    Sampling site and meteorology

The $A^2BC$ was carried out at the national weather station located in XT (37.18º N,
114.36º E) from 1 May to 15 June of 2016. This site is situated in southern Heibei
Province, located in the central part of the NCP and to the east of Taihang Mountains
(Fig. 1a). This region is heavily populated, urbanized, and industrialized. The major
industrial manufacturers include coal-based power plants, steel and iron works,
glassworks, and cement mills. The weak diffusion conditions and heavy industrial
emissions lead to exceptionally high concentrations of particulate matter (PM) with
diameter less than 10 μm ($PM_{10}$) and 2.5 μm ($PM_{2.5}$), as well as gas pollutants such as
sulfur dioxide ($SO_2$), and nitrogen oxides ($NO_x$) during the frequent occurring haze
episodes in this region (Wang et al., 2014; Fu et al., 2014). Figure 1b shows the mean
distribution of $SO_2$ concentrations from May of 2012 to 2016, confirming that the
measurement site is located in the pollution center of this region.
Time series of meteorological variables measured at the weather station are
shown in Fig. S1. This site is heavily affected by the mountain-valley wind, showing a
prevailing southeasterly wind during the day and a northwesterly wind at night (Fig.
S1 and Fig. S2). There was almost no precipitation during the study period. The
ambient temperature ($T$) and relative humidity (RH) time series show opposing trends.





Campaign-mean values of $T$ and RH are 21.9 °C and 51.6 %, respectively.

**2.2.    Instrumentation and operation**

**2.2.1.    Aerosol hygroscopicity measurements**

The hygroscopicity tandem differential mobility analyzer (H-TDMA) used in this
study has been described in detail by others (Tan et al., 2013; Y. Wang et al., 2017).
Briefly, ambient aerosols are first dried and neutralized by a Nafion dryer and a soft
X-ray charger. A differential mobility analyzer (DMA$_1$, model 3081L, TSI Inc.) is
used to select monodispersed particles of a certain diameter ($D_{p0}$). The monodisperses
particles are then passed through a Nafion humidifier with a controlled higher RH and
are humidified. A second DMA (DMA$_2$, same model as the DMA$_1$) and a water-based
condensation particle counter (WCPC, model 3787, TSI Inc.) are used to measure the
number size distribution of the humidified particles. The DMA$_1$ and WCPC can also
be connected directly to measure the 10–400 nm particle number size distribution
(PNSD). In this study, the dry diameters selected by the DMA$_1$ are 40, 80, 110, 150,
and 200 nm. The humidified RH is set to 85 %.
The hygroscopic growth factor (GF) is defined as the ratio of the humidified
diameter at a given RH to the dry diameter:

$$GF = \frac{D_p(RH)}{D_{p0}}, \tag{1}$$

where $D_p(RH)$ is the particle diameter at the given RH and $D_{p0}$ is the dry diameter
selected by the DMA$_1$. The measured distribution function versus GF (GF-MDF) can
be calculated with WCPC data downstream from the DMA$_1$ and DMA$_2$. The GF



probability density function is then retrieved using the TDMAFIT algorithm
(Stolzenburg and McMurry, 1988, 2008).

### 2.2.2. Aerosol chemical composition measurements

The Aerosol Chemical Speciation Monitor (ACSM) was deployed to measure the
non-refractory submicron aerosol (NR-PM$_1$) species (sulfate, nitrate, ammonium,
chloride, and organics) in real-time. A PM$_{2.5}$ URG cyclone (model URG-2000-30ED)
was installed in the front of the sampling inlet to remove coarse particles (> 2.5 μm in
diameter). Before sampling into the ACSM, aerosol particles were dried (below 40 %
RH) by a silica gel diffusion dryer. In addition, the ACSM was calibrated routinely
with pure ammonium nitrate to determine its ionization efficiency. More detailed
descriptions about the ACSM are given by Ng et al., (2011) and Sun et al., (2012). A
positive matrix factor analysis is used to analyze the organic spectral matrices
according to Ulbrich et al., (2009). Three factors, i.e., hydrocarbon-like OA (HOA),
cooking OA (COA), and oxygenated OA (OOA), are chosen as the ACSM dataset.
HOA and COA are both anthropogenic primary organic aerosols (POA) while OOA is
the secondary organic aerosol (SOA).
The ACSM does not detect refractory material such as BC, so a seven-wavelength
aethalometer (AE-33, Magee Scientific Corp.) was used to measure the BC mass
concentration of BC particles with diameters < 1.0 μm (BC PM$_1$). Mineral dust and
sea salt are the other refractory species, but they typically exist in the coarse mode and
make negligible contributions to PM$_1$ (Juranyi et al., 2010; Meng et al., 2014).




### 2.2.3. Aerosol size distribution and CCN measurements

The aerosol particle number size distribution (15–685 nm) was measured by a scanning mobility particle sizer (SMPS) that was equipped with a long DMA (model 3081L, TSI Inc.) and a condensation particle counter (CPC, model 3775, TSI Inc.). A single-column continuous-flow stream-wise thermal-gradient cloud condensation nuclei counter (CCNC-100, DMT Inc.) was applied to measure the bulk CCN number concentration. Five SS levels, i.e., 0.07, 0.1, 0.2, 0.4, and 0.8 %, were set in the CCNC and the running time was 10 min for each SS level. The SS in the CCNC are calibrated with pure ammonium sulfate (Rose et al., 2008) before and after the measurement campaign. The corrected SS levels are 0.11, 0.13, 0.22, 0.40, and 0.75 %, respectively.

The aerosol activation ratio (AR) at a certain SS is calculated as $N_{CCN}$ divided by the total particle number concentration in the 15–685 nm range ($N_{15–685\ nm}$), i.e., AR = $N_{CCN}$ / $N_{15–685\ nm}$. The particle number concentration below 15 nm is not measured by the SMPS, but this does not affect the calculated $N_{CCN}$ because the activation critical diameter is always larger than 15 nm at these SS levels (Zhang et al., 2014). Aerosol particles with diameters larger than 685 nm are also not detected by the SMPS. These larger particles will always act as CCN due to their larger dry sizes. However, the number concentration above 685 nm in the atmosphere is always negligible (Juranyi et al., 2010).



### 2.2.4. Other measurements

In this study, a micro-pulse lidar (MPL-4B, Sigmaspace Corp.) was used to study
the evolution of the planetary boundary layer (PBL). The pulse repetition rate of the
MPL was 2.5 kHz at a visible wavelength of 532 nm. The peak value of the optical
energy of the laser beam was 8 μJ. The pulse duration ranged from 10 to 100 ns, and
the pulse interval was set to 200 ns, corresponding to a spatial resolution of 30 m. The
MPL-retrieved PBL height is the altitude where a sudden decrease in the scattering
coefficient occurs (Brooks, 2003; Quan et al., 2013).
Trace gas analyzers were used to measure the gaseous species of ozone ($O_3$) and
$SO_2$. $SO_2$ was measured using an $SO_2$ analyzer with a fluorescence cell (Ecotech
model 9850A) and $O_3$ was measured using an $O_3$ analyzer (Ecotech model 9810B)
with ultraviolet absorption technology. More detailed descriptions about the trace gas
analyzers are given by Zhu et al., (2016).

### 3. Theory

### 3.1. Hygroscopicity parameter

To link hygroscopicity measurements below and above water vapor saturation,
the Köhler theory (Köhler, 1936) is parameterized using the hygroscopicity parameter
$\kappa$ (Petters and Kreidenweis, 2007). This is known as the $\kappa$-Köhler theory. According
to the theory, the equilibrium equation over a solution droplet at a saturation ratio
$S(D)$ is



$$S(D) = \frac{D^3 - D_d^3}{D^3 - D_d^3(1-\kappa)} \exp\left(\frac{4\sigma_{s/a}M_w}{RT\rho_w D}\right) \quad , \tag{2}$$

where $D$ and $D_d$ are the wet and dry droplet diameters, respectively, $\sigma_{s/a}$ is the
surface tension coefficient, $M_w$ is the mole mass of water, $R$ is the universal gas
constant, $T$ is the temperature, and $\rho_w$ is the density of water.

Below the water vapor saturation, $S(D)$ is RH, $D$ is $D_p(\text{RH})$, and $D_d$ is $D_{p0}$

in Eq. (1). The κ parameter is then calculated using H-TDMA data according to Eq. (1)
and Eq. (2):
$$\kappa_{gf} = (GF^3 - 1) \cdot \left[\frac{1}{RH}\exp\left(\frac{4\sigma_{s/a}M_w}{RT\rho_w D_d GF}\right) - 1\right] \quad . \tag{3}$$

For a multicomponent particle, the Zdanovskii–Stokes–Robinson (ZSR) mixing

rule (Stokes and Robinson, 1966) can also estimate κ using chemical composition
data:
$$\kappa_{chem} = \sum_i \varepsilon_i \kappa_i, \tag{4}$$

where $\varepsilon_i$ and $\kappa_i$ are the volume fraction and hygroscopicity parameter for the $i$th
chemical component. The ACSM provides the mass concentrations of inorganic ions
and organics. A simplified ion-pairing scheme such as that described by Gysel et al.,
(2007) was applied to convert ion mass concentrations to mass concentrations of their
corresponding inorganic salts (see Table S1 in the supplement). Table S1 also lists κ
and the gravimetric density of each individual component. In the following
discussions, $\kappa_{gf}$ and $\kappa_{chem}$ denote the hygroscopicity parameters derived from
H-TDMA measurements and estimated using the ZSR mixing rule, respectively.



**3.2. CCN estimation**


The critical supersaturation ($s_c$, $s_c = S_c-1$) for a dry diameter ($D_d$) of a particle with
hygroscopicity $\kappa$ is calculated from the maximum of the $\kappa$-Köhler curve (Eq. 1)
(Petters and Kreidenweis, 2007). The $s_c$-$D_d$ relationship is then established. According
to this relationship, the critical diameter ($D_{0,\mathrm{crit}}$) can be calculated using the estimated
$\kappa_{\mathrm{chem}}$ (Eq. 4) at the SS set in the CCNC. All particles larger than $D_{0,\mathrm{crit}}$ will activate
as CCN, assuming that aerosols are internally mixed. Then the CCN number
concentration can be estimated from the integral of the aerosol size distribution
provided by the SMPS from $D_{0,\mathrm{crit}}$ to the largest measured size.

**4. Results and discussion**


**4.1. Overview**


Figures 2 and 3 show the time series of the main aerosol properties during the
field experiment. The PNSD changes dramatically (Fig. 2a) and the aerosol number
concentration in the 15–50 nm range ($N_{15\text{–}50\,\mathrm{nm}}$) increases sharply in the morning
almost every day (Fig. 2b). The time series of the mean diameter ($D_\mathrm{p}$) of particles also
shows that a growth process occurs after the sharp increase in $N_{15\text{–}50\,\mathrm{nm}}$. All these
phenomena suggest that new particle formation (NPF) events occurred frequently at
XT during the field experiment (Kulmala et al., 2012; Y. Li et al., 2017). This is likely
related to the high concentration of gas precursors mainly from local emissions. High
emissions of $SO_2$ and volatile organic compounds associated with the high oxidation



capacity in a polluted atmosphere make NPF events occur more frequently in north
China (Z. Wang et al., 2017).
Figure 2c-d shows the time series of the probability density function of $\kappa_{gf}$
($\kappa$-PDF) for 40 nm and 150 nm particles, respectively. In general, mono-modal
$\kappa$-PDFs are observed. This is different from $\kappa$-PDFs at other sites in China where bi-
and tri-modal distributions are dominant (Liu et al., 2011; Ye et al., 2013; Jiang et al.,
2016; S.L. Zhang et al., 2016; Y. Wang et al., 2017). This is due to differences in the
aerosol mixing state, which will be discussed in section 4.2.
The bulk mass concentrations of organics, sulfate, nitrate, ammonium, and
chloride measured by the ACSM are shown in Fig. 3a, along with the BC mass
concentration measured with the AE-33. Organics and sulfate are the dominant
chemical species with mass fractions in $PM_1$ of 39.1 % and 24.7 %, respectively.
Figure 3b-c shows the volume fractions of paired chemical compositions and the
hygroscopicity parameter derived from $\kappa_{chem}$, respectively. The average volume
fraction of inorganics $((NH_4)_2SO_4+NH_4HSO_4+H_2SO_4+NH_4NO_4)$ is similar to that of
organics (POA+SOA), but their volume fractions change diurnally. The volume
fraction of inorganics increases during daytime while the volume fraction of organics
decreases. In addition, SOA is the dominant contributor to OA, accounting for ~69 %
of the organics volume. This shows that photochemical reactions were strong at XT
during the field experiment (Huang et al., 2014). The mean $\kappa_{chem}$ in Fig. 3c is 0.31
with values ranging from 0.20 to 0.40. The trend in $\kappa_{chem}$ is similar to that of the
volume fraction of inorganics, suggesting that inorganics played a key role when it



comes to $\kappa_{\text{chem}}$ (Wu et al., 2016).

## 4.2.  Aerosol mixing state and hygroscopicity

The average probability density functions of $\kappa_{\text{gf}}$ ($\kappa$-PDF) for different particle

sizes derived from H-TDMA data are shown in Fig. 4. For all particle sizes considered,
$\kappa_{\text{gf}}$ ranges from 0 to 0.8 and the $\kappa$-PDF patterns are similar, suggesting that the
hygroscopic compounds in different particle size mode were similar at XT during the
field experiment. In general, $\kappa$-PDF patterns show only one hydrophilic mode with
weak hydrophobic modes occasionally appearing at night when photochemical
reactions are weak (Fig. S3). This is different from what has been reported at other
sites in China (Liu et al., 2011; Ye et al., 2013; Jiang et al., 2016; Zhang et al., 2016;
Y. Wang et al., 2017) where the $\kappa$-PDF patterns always show bi- or tri-modal
distributions. Based on previous studies (Liu et al., 2011; Y. Wang et al., 2017),
ambient aerosols can be classified into three groups according to their $\kappa_{\text{gf}}$ values:

— nearly hydrophobic (NH): $\kappa_{\text{gf}} < 0.1$

— less hygroscopic (LH): $0.1 \leq \kappa_{\text{gf}} < 0.2$

— more hygroscopic (MH): $0.2 \leq \kappa_{\text{gf}}$

Table 1 gives the number fractions of each group for different particle sizes. The MH
group dominates all particle sizes. The number fractions of the NH and LH groups are
less than 6.0 % each. However, the volume fractions of hydrophobic BC and
low-hygroscopic organics (where $\kappa_{\text{BC}}$ is approximately zero and $\kappa_{\text{organic}}$ is
typically less than 0.1) are ~10.1 % and 47.4 % according to chemical composition



295 measurements (Fig. 3b). This suggests that the particles were highly aged and

296 internally mixed at XT during the field experiment. Coating of sulfates and secondary

297 organics during the aging process changes the structure of BC and makes it grow,

298 which can significantly enhance the hygroscopicity of particles (e.g., Zhang et al.,

299 2008; Jimenez et al., 2009; Tritscher et al., 2011; Guo et al., 2016). In addition, the

300 observed unimodal distribution of $\kappa$-PDF also suggests the highly internally mixed

301 state of the particles (Swietlicki et al., 2008).

302  Figure 5 shows the average size-resolved $\kappa_{gf}$ derived from H-TDMA data at XT

303 and at other sites in China. At XT, $\kappa_{gf}$ for different particle sizes are larger in the

304 daytime than at night and the difference between daytime and nighttime decreases

305 with increasing particle size. This suggests that the impact of photochemical reactions

306 on aerosol hygroscopicity was strong and that the effect was weaker with increasing

307 particle size.

308  The magnitude of $\kappa_{gf}$ is larger at XT than at other sites of China. In particular,

309 the magnitude of $\kappa_{gf}$ is much larger at XT than at sites in the northern part of the

310 NCP, i.e., Beijing, Wuqing, and Xianghe. The lower $\kappa_{gf}$ in the urban area of Beijing

311 is likely related to the more severe traffic emissions (Ye et al., 2013; Wu et al., 2016).

312 Wuqing and Xianghe are located in the suburban area between the two megacities of

313 Beijing and Tianjin and are simultaneously affected by traffic and industrial emissions.

314 The magnitude of $\kappa_{gf}$ at these two sites are higher than at Beijing but lower than at

315 XT. Although XT is located far away from these megacities, it is situated in the

316 industrial center of the NCP, so the higher concentrations ofprecursors and strong





photo chemical reactions make the particles more internally mixed and highly aged.
This is why $\kappa_{\mathrm{gf}}$ in XT is larger than at other sites. This suggests that the
hygroscopicity of particles from different emissions and chemical processes differ in
NCP. In addition, 40 nm particles are always more hygroscopic than 80 nm particles
at XT, especially in the daytime, which is also different from other sites. This is likely
because the coating effect of sulfates and secondary organics is more significant on
smaller particles (Tritscher et al., 2011; Guo et al., 2016). Furthermore, since the field
measurements took place in a local with heavy industrial activities, it is possible that
amine contributes significantly to the hygroscopicity of 40 nm particles. Several
studies have shown that amine compounds in aerosol phase can be hygroscopic,
sometimes at even low RH (e.g. Qiu and Zhang, 2012; Chu et al., 2015;
Gomez-Hernandez et al., 2016). .
**4.3.  Diurnal variations in aerosol properties**
**4.3.1.  Diurnal variations in aerosol number and mass concentrations**

Figure 6a shows the diurnal variation in MPL-derived PBL height. PBL height

can be determined at the altitude where a sudden decrease in the scattering coefficient
occurs from the MPL data (Cohn and Angevine, 2000; Brooks, 2003). Note that the
retrieved PBL height is only valid from 07:00 local time (LT) to 19:00 LT (Quan et al.,
2013). The retrieved PBL height at night is not accurate because of the likely
influence of residual aerosols within the nocturnal PBL. The evolution of PBL height
from 07:00 LT to 19:00 LT is sufficient to analyze its link with the change in aerosol





number and mass concentrations during the daytime. Figure 6b shows diurnal
variations in aerosol number and mass concentrations in the 15–685 nm range ($N_{15–685}$
$_{nm}$ and $PM_{15–685\ nm}$, respectively). Variations in the $N_{15–685\ nm}$ and $PM_{15–685\ nm}$ trends
oppose each other. From 08:00 LT to 14:00 LT, the PBL height lifts from ~0.5 km to
~0.6 km, while $PM_{15–685\ nm}$ decreases from ~24 μg m$^{-3}$ to ~19 μg m$^{-3}$ although there is
a slight increase at the beginning of the period. This suggests the important effect of
PBL evolution on $PM_{15–685\ nm}$. However, $N_{15–685\ nm}$ sharply increases from ~7600 cm$^{-3}$
at 07:00 LT to ~13000 cm$^{-3}$ at 13:00 LT. This is related to the sudden burst of
nucleation mode particles (< 100 nm) when NPF events occurred. Newly formed fine
particles contribute little to $PM_{15–685\ nm}$. In the evening, $PM_{15–685\ nm}$ increases
gradually while $N_{15–685\ nm}$ decreases. This is attributed to the declining trend in the
nocturnal PBL and particle coagulation and growth. In other words, the evolution of
the PBL plays a dominant role on the aerosol mass concentration, while particle
formation and growth has a greater influence on the variation in aerosol number
concentration.
**4.3.2.  Diurnal variation in aerosol hygroscopicity**

Figure 6c shows diurnal variations in $\kappa_{gf}$ and $\kappa_{chem}$. All sized $\kappa_{gf}$ increases

beginning from the NPF event, especially for the 40 nm particles. The increase of $\kappa_{gf}$
in the morning was synchronous with the particle number concentration ($N_{15–685\ nm}$)
but not with the PBL height, further suggesting the impact of photochemical reactions
on aerosol hygroscopicity. The $\kappa_{gf}$ for 40 nm particles increases from ~0.32 at 07:00





LT to ~0.44 at 15:00 LT, and approaches the $\kappa$ of pure ammonium sulfate
[$\kappa_{gf,(NH_4)_2SO_4}$= 0.48 (Wu et al., 2016)]. This suggests that a large amount of sulfates
were produced through the photochemical reactions of precursors. This can be verified
from the diurnal variation in chemical composition (Fig. 6d), which shows that the
mass fraction of sulfate increased during the daytime while the mass fraction of
organics (POA or SOA) varied more weakly. The diurnal variation in $SO_2$ precursor
also shows this (Fig. S4). The diurnal variation in $\kappa_{gf}$ for 80–200 nm particles differs
from that of 40 nm particles. The differences in $\kappa_{gf}$ between 80–200 nm particles in
the early morning are large but gradually decrease as the sun rises. After 11:00 LT, the
$\kappa_{gf}$ for 80–200 nm particles are similar but lower than that of 40 nm particles,
suggesting that the enhanced hygroscopicity in the 80–200 nm particles was likely
caused by the condensation of sulfates and secondary organics while that of the 40 nm
particles was caused by the growth of the new particles.

Figure 6c also shows that the $\kappa_{chem}$ for $PM_1$ is lower than the $\kappa_{gf}$ for 40–200

nm particles and has a weaker diurnal variation. Two reasons may explain this: (1) the
bulk chemical compositions of $PM_1$ and of 40–200 nm particles differ greatly and (2)
the ZSR model cannot describe the impact of condensation on aerosol hygroscopicity
very well. During the daytime, the condensation of sulfuric acid on organics or BC
greatly enhances their hygroscopicity (Zhang et al., 2008; Zhang et al., 2009). Cruz
and Pandis (2000) have shown that the measured $\kappa_{gf}$ of internally mixed
$(NH_4)_2SO_4$-organic aerosols is larger than the predicted $\kappa_{chem}$ based on the ZSR
model.



In summary, the ample supply of effluent $SO_2$ and VOCs provided sufficient
precursors for the strong photochemical reactions at XT during the field experiment,
and the produced sulfates and the condensation of sulfuric acid enhanced aerosol
hygroscopicity, especially during the day. This also suggests that the observed
frequent NPF events were mainly induced by the oxidation of precursors.
**4.3.3.  Diurnal variation in CCN number concentration and activation ratio**
Figure 7a shows the diurnal variations in $N_{CCN}$ and AR at different SS. In the
morning, $N_{CCN}$ first decreases then increases while AR shows the opposite trend. This
is related to the evolution of the PBL and NPF events. At the initial stage of an NPF
event, the newly formed particles are less than 15 nm in size, which is below the
detection limit of the SMPS. As a result, $N_{15–685\ nm}$ decreases (Fig. 6b) as the PBL lifts
and $N_{CCN}$ also decreases. However, the mixing of aged particles within the PBL makes
the particle size (Fig. 7b) and AR increase slightly. With condensation and the growth
of new particles, fine particles detected by the SMPS increase rapidly but a portion of
them cannot be activated because their smaller size. So $N_{CCN}$ increases but AR
decreases from 08:00 LT to 14:00 LT. In the afternoon and evening, $N_{CCN}$ and AR
increase slightly with the increase in particle size (Fig. 7b). However, these trends
become weaker as SS decreases, likely because the critical diameter is larger at low
SS and the influence of aerosol size distribution on $N_{CCN}$ and AR is relatively weaker.
This demonstrates that the particle size is the most important factor influencing the
aerosol activation ability and the CCN number concentration, especially at larger SS





levels.
**4.4.   CCN estimation from chemical composition data**

The three main factors influencing CCN activation are particle size, mixing state,

and chemical composition. As discussed in the above sections, particles are highly
internally mixed at XT and particle size has a great influence on $N_{CCN}$. In this section,
a CCN closure study is conducted and the impact of chemical composition on $N_{CCN}$ is
discussed. Figure 8a shows estimated $N_{CCN}$ as a function of measured $N_{CCN}$ using
real-time $\kappa_{chem}$. The estimated $N_{CCN}$ correlates well with measurements ($R^2 \geqslant 0.85$)
but is generally overestimated. The slope of each linearly fitted line is greater than
1.10 and increases with increasing SS. In addition, the relative deviation (RD)
increases from 16.2 % to 25.2 % as SS increases from 0.13 % to 0.75 %, suggesting
that estimates become worse at larger SS. The overestimation of $N_{CCN}$ is mainly
caused by large measurement uncertainties of CCNC: (1) the temperature or high flow
rates in the CCNC may not allow enough time for particles to reach sizes large
enough to be counted by the OPC at the exit of the CCN chamber (Lance et al., 2006;
Cubison et al., 2008) and (2) in high particle number concentration environments,
water depletion in the CCNC may reduce the counting rate of the CCNC (Deng et al.,
2011). These uncertainties make measured $N_{CCN}$ lower than the actual $N_{CCN}$. At larger
SS, those activated aerosols in the cloud chamber of CCNC are greater in number and
smaller in size, so the impact of these uncertainties is greater. Another discussion
about this problem can be found in the supplement (Fig. S5).



Figure 8b shows estimated $N_{CCN}$ using the mean value for $\kappa_{chem}$ ($\kappa_{chem} = 0.31$).
Compared with results using real-time values for $\kappa_{chem}$, the fit parameters and RD
change slightly, suggesting that the effect of chemical composition on $N_{CCN}$ is weaker
relative to the particle size. The sensitivity of estimated $N_{CCN}$ to the variability in
chemical composition ($\kappa_{chem}$) is further investigated (Fig. 9). In this figure, the
variability of the equipotential lines in RD suggests that the sensitivity of $N_{CCN}$ is
strongly time dependent. This is attributed to the variability of the shape of the aerosol
size distribution (Juranyi et al., 2010). The sensitivity of $N_{CCN}$ to chemical
composition ($\kappa_{chem}$) becomes weaker with increasing SS, suggesting that chemical
composition becomes less important in $N_{CCN}$ estimates at larger SS. In addition, the
RD is always less than 10 % when estimating $N_{CCN}$ using the mean value of $\kappa_{chem}$,
suggesting that $\kappa = 0.31$ is a good proxy for chemical composition when estimating
$N_{CCN}$ at XT.
In summary, particle size is the most important factor influencing the aerosol
activation ability at XT, especially at larger SS levels. The mixing state and chemical
composition were not as important when estimating $N_{CCN}$ because the particles were
highly aged and internally mixed at XT during the field experiment, and aerosol
hygroscopicity was not sensitive to estimates of $N_{CCN}$.
**5.    Summary and conclusions**
The Atmosphere-Aerosol-Boundary Layer-Cloud (A$^2$BC) Interaction Joint
Experiment was carried out at a polluted site located in the central North China Plain



(NCP) from 1 May to 15 June of 2016. The aerosol hygroscopicity, mixing state and
CCN activity at the site Xingtai (XT) were investigated in this study.

In general, the probability density function of the hygroscopicity parameter

($\kappa$-PDF) for 40–200 nm particles is a unimodal distribution, which is different from
distributions at other sites in China. Particles of all sizes cover a large range of $\kappa_{gf}$
(mostly from 0 to 0.8) and show similar $\kappa$-PDF patterns, suggesting that the
hygroscopic compounds in these particles from 40 nm to 200 nm were similar at XT.
The $\kappa$-PDF patterns also suggests that the particles were highly aged and internally
mixed at XT during the field experiment. This is likely related to strong
photochemical reactions.

The mean $\kappa_{gf}$ for different particle sizes are larger in the daytime than at night.

Daytime and nighttime $\kappa_{gf}$ differences decrease with increasing particle size. This
illustrates that the impact of photochemical reactions on aerosol hygroscopicity was
strong and that the effect became weaker as particle sizes increased. The enhanced
hygroscopicity of 40–200 nm particles was likely caused by the coating of sulfates or
secondary organics while the effect was more significant for  40 nm particles.
Compared with other sites in China, the aerosol hygroscopicity was much larger at XT
because of the strong photochemical reactions and the sufficient precursors. The
comparison also shows that the hygroscopicity of particles from different emissions
and chemical processes differed.

New particle formation events occurred frequently at XT during the field

experiment. The evolution of the planetary boundary layer (PBL) played a dominant



role on the aerosol mass concentration, while particle formation and growth had a
greater influence on the variation in the aerosol number concentration. Particle size
was the most important factor influencing the aerosol activation ability and the CCN
number concentration at XT during the field experiment, especially at larger
supersaturations (SS). Although the estimated $N_{CCN}$ correlated well with
measurements ($R^2 \geqslant 0.85$), $N_{CCN}$ was overestimated because of measurement
uncertainties. The effect of chemical composition on $N_{CCN}$ was weaker relative to the
particle size. Sensitivity tests show that the impact of chemical composition on $N_{CCN}$
became weaker as SS increased, suggesting that the effect of chemical composition on
$N_{CCN}$ estimates is less important at larger SS. The value $\kappa = 0.31$ is a good proxy for
chemical composition when estimating $N_{CCN}$ for the model at XT.
Our results show that aerosol properties in the middle of the NCP differ from
those in the northern part of the NCP and other regions in China. This is because there
are more industrial emissions in the central NCP. The plenitude of gas precursors and
strong photochemical reactions at XT make aerosol properties there different from
those at sites under other polluted conditions. More field measurements on
gas-particle transformation and aerosol properties in this region are needed, which
would be meaningful for studying the haze formation mechanism and climate change
in the NCP.

*Data availability*. The data used in the study are available from the first author upon
request (wang.yuying@mail.bnu.edu.cn).

*Competing interests*. The authors declare that they have no conflict of interest.





*Author contribution*. Z.L. and Y.W. designed the experiment, Y.W., Y.Z., and W.D.
carried it out and analyzed the data, other co-authors participated in science discussions
and suggested analyses. Y.W. prepared the manuscript with contributions from all
co-authors.
*Acknowledgements*. This work was funded by the National Natural Science
Foundation of China (NSFC) research projects (grant no. 91544217, 41675141), the
National Basic Research Program of China "973" (grant no. 2013CB955801), and the
China Scholarship Council (award no. 201706040194). We also thank all participants
in the field campaign for their tireless work and cooperation.

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

**Table 1.** The number fractions of different hygroscopic groups for different particle
sizes.

|     | 40 nm | 80 nm | 110 nm | 150 nm | 200 nm |
|-----|-------|-------|--------|--------|--------|
| NH  | 5.1 % | 5.0 % | 5.1 %  | 5.0 %  | 5.7 %  |
| LH  | 4.8 % | 4.2 % | 4.3 %  | 4.7 %  | 5.1 %  |
| MH  | 90.1 %| 90.8 %| 90.6 % | 90.3 % | 89.2 % |




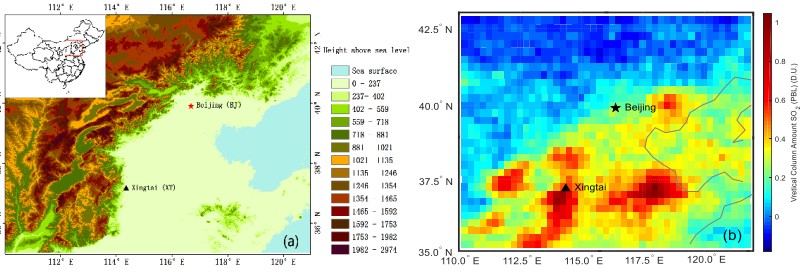





**Figure 1. (a)** Map showing the locations of the sampling sites and **(b)** the distribution
of mean SO₂ concentrations from May of 2012 to 2016.

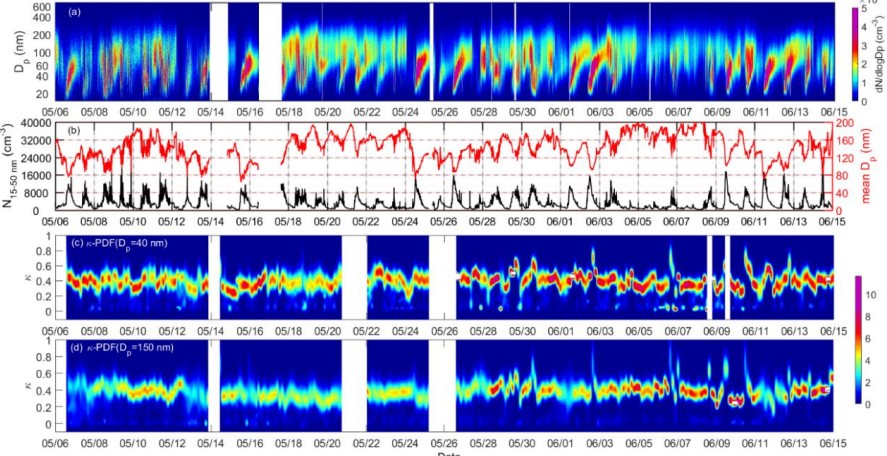


**Figure 2.** The time series of **(a)** particle number size distribution (PNSD), **(b)** aerosol
number concentration in the 15–50 nm range ($N_{15-50\,nm}$) and the mean diameter ($D_p$),
**(c)** the probability density function of $\kappa_{gf}$ ($\kappa$-PDF) for 40 nm and **(d)** 150 nm
particles from 6 May to 15 June of 2016.



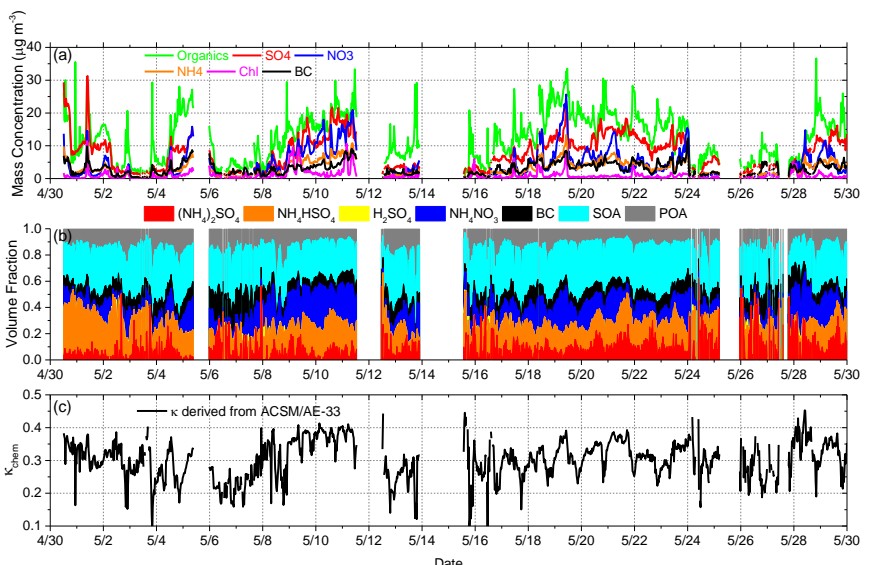


**Figure 3.** Time series of (**a**) the bulk mass concentration of aerosol species in PM$_1$, (**b**)

the volume fractions of POA, SOA, BC, and inorganics with the simplified ion

pairing scheme, and (**c**) the hygroscopicity parameter derived from the chemical

compositions ($\kappa_{chem}$).










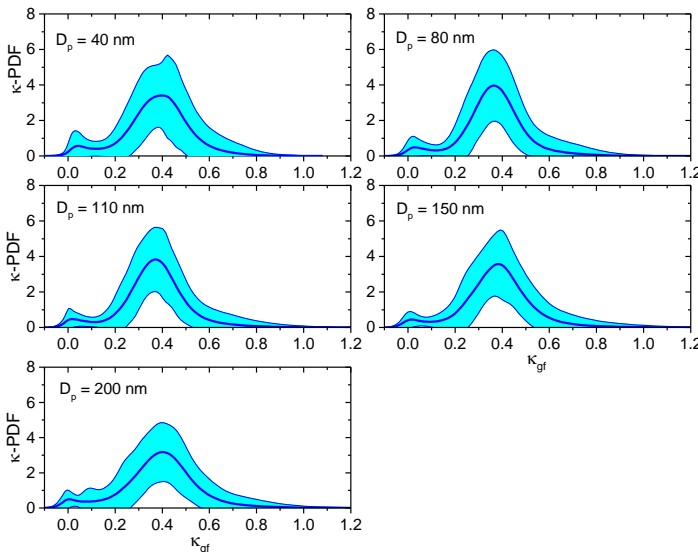


**Figure 4.** Mean probability density functions of $\kappa_{gf}$ ($\kappa$-PDF) for different particle

sizes and their standard deviations (shaded areas) derived from H-TDMA data and

measured at RH = 85 %.


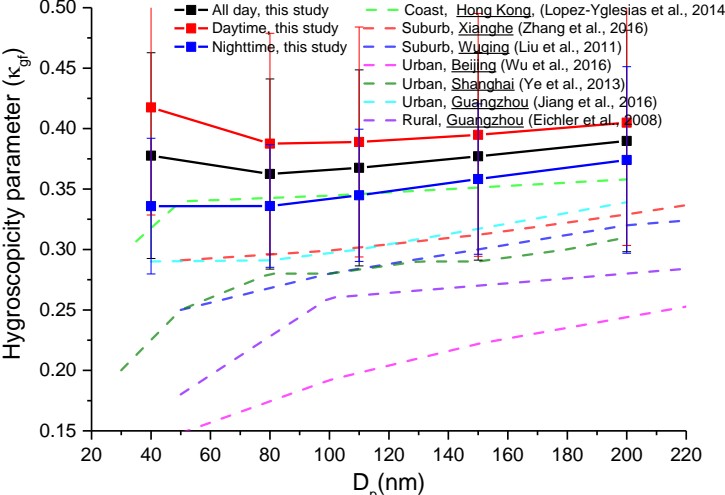


**Figure 5.** Size-resolved aerosol hygroscopicity parameter ($\kappa_{gf}$) derived from

H-TDMA data at XT and at other sites in China.


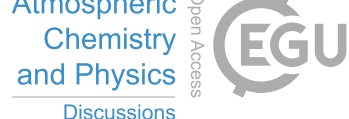

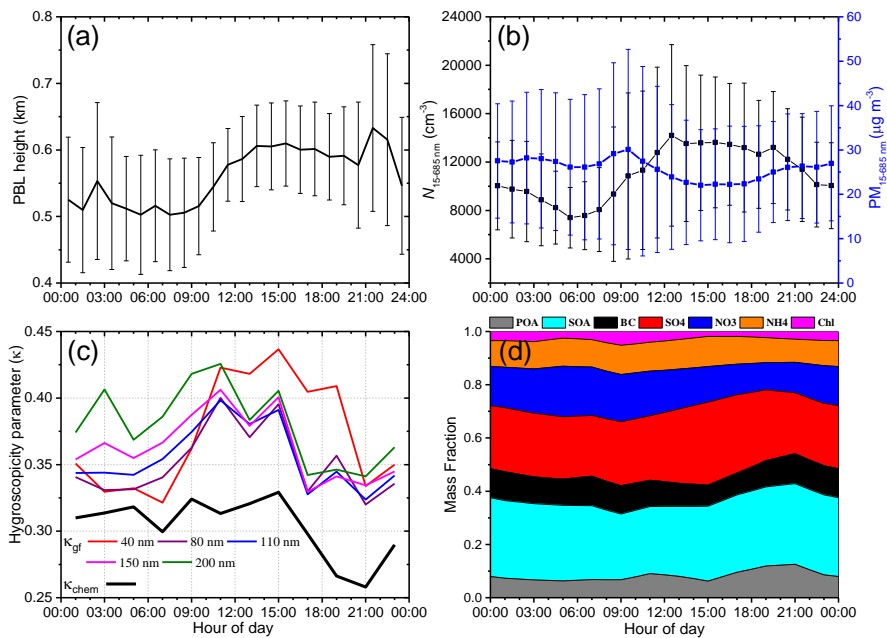

**Figure 6.** Diurnal variations in (**a**) planetary boundary layer (PBL) height retrieved

from the MPL, (**b**) aerosol number and mass concentrations in the 15–685 nm range

($N_{15–685\ nm}$ and $PM_{15–685\ nm}$, respectively) derived from the SMPS (an aerosol density

of 1.6 g cm$^{-3}$ is assumed), (**c**) the hygroscopicity parameter derived from the

hygroscopic growth factor ($\kappa_{gf}$) and predicted from the bulk chemical composition

($\kappa_{chem}$), and (**d**) the mass fractions of different species.

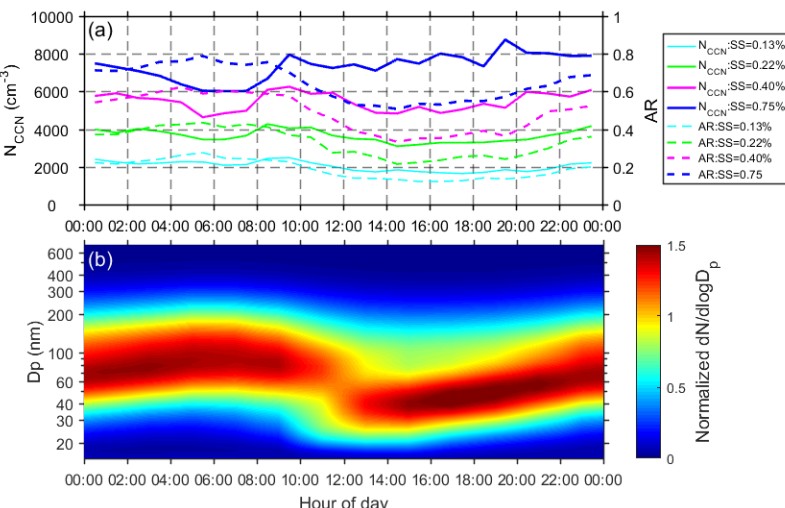


**Figure 7.** Diurnal variations in (a) CCN number concentration ($N_{CCN}$) and activation

ratio (AR), and (b) the normalized aerosol size distribution in the 15–685 nm particle

size range.


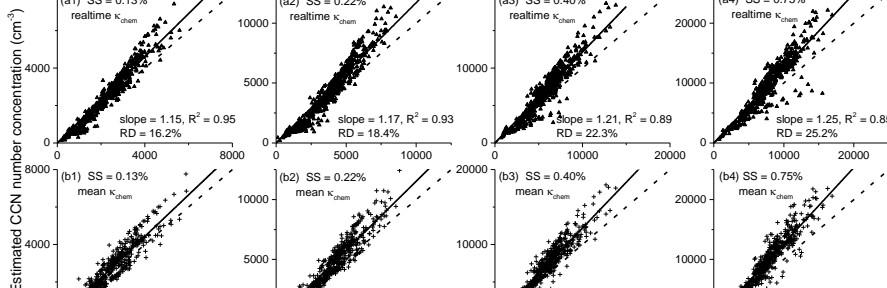


**Figure 8.** Estimated versus measured CCN number concentration for ambient

aerosols at four different supersaturation levels. The $N_{CCN}$ is estimated based on

κ-Köhler theory, using the real-time $\kappa_{chem}$ (**a1-a4**) and the mean $\kappa_{chem}$ (**b1-b4**).

The slope and correlation coefficient ($R^2$) of the linear regression, and the relative



deviation of estimated $N_{CCN}$ (RD = $|N_{CCN\_estimated} - N_{CCN\_measured}| / N_{CCN\_measured}$) are
shown in each panel. The regression line is overlaid on the measurements (solid line)
and the dashed line is the 1:1 line.


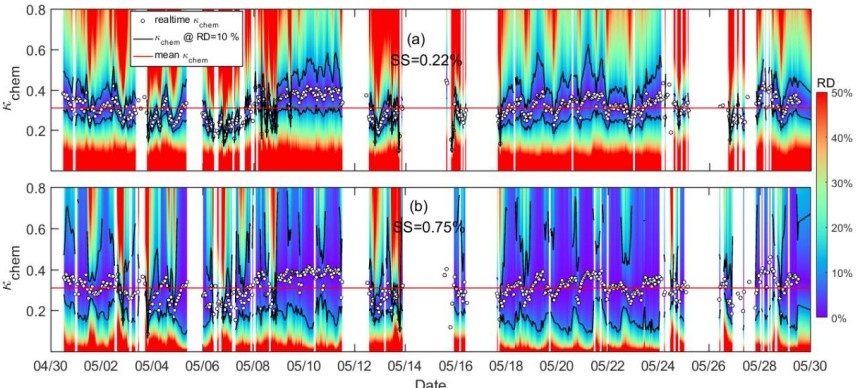


**Figure 9.** Sensitivity of $N_{CCN}$ estimates to $\kappa_{chem}$ as a function of time at (a) SS =
0.22 % and (b) SS = 0.75 %. The color scale indicates the relative deviation (RD) of
the CCN estimates using the $\kappa_{chem}$ value shown on the ordinate. In each panel, open
circles show the real-time $\kappa_{chem}$. Note that RD is by definition zero at these points.
The black line is κ at RD = 10 % and the red line is the mean value for $\kappa_{chem}$ (0.31).
Figure S6 in the supplement shows the same plots but for SS = 0.13 % and 0.40 %.