# Peer review of "Characterization of aerosol hygroscopicity, mixing state, and 1 CCN activity at a suburban site in the central North China Plain 2 3 Yuying Wang1, Zhanqing Li1, Yingjie Zhang2, Wei Du2,3, Fang Zhang1, Haobo Tan4, 4 5 Hanbing Xu5, Tianyi Fan1, Xiaoai Jin1, Xinxin Fan1, Zipeng Dong1, Qiuyan Wang6, Yele 6 Sun2,3 7 8 9 1College of Global Change and Earth System Science, Beijing Normal University, Beij"

_Atmospheric Chemistry and Physics, 2017_

## Referee Comment (RC1) · Anonymous Referee #2 · 22 Jan 2018

This study reported field measurements results of the chemical speciation, hygroscopicity and CCN properties of ambient particles at a suburban site in the central North China Plain (NCP). The probability density function of the hygroscopicity parameter kappa-PDF was then derived from measurement data and showed only a singular hygrophilic mode which was very different from profiles observed in other regions of the NCP (that were normally bi- or tri-modal). Among the possible factors affecting aerosol hygroscopicity (the mixing state, chemical composition and particle size), particle size was identified as the key factor influencing the particle CCN activation.

This study explored the aerosol microphysical properties in a region that was not previously studied and the results can be useful when compared with existing data to understand aerosol aging and its impact on particle microphysics and the climate. The topic is relevant to the scope of the journal of Atmospheric Chemistry and Physics and should be considered for publication.

(1) The tile may be changed to "Characterization of aerosol hygroscopicity, mixing state, and CCN activity at a suburban site in the central North China Plain" to reflect the unique location of this study.

(2) Section 4.1, Lines 317-321: While it's been shown that aging of BC will enhance its hygroscopicity and CCN activation, the actual determination of the GF of aged BC could be challenging (see, for example, Torsten et al., Environ. Res. Lett., 2011) as the DMA mobility size change may be marginal. A few chambers studies on soot SOA from anthropogenic VOCs may provide some insights here (Guo et al., ES&T, 2016 and Qiu et al., ES&T, 2012). In general, the knowledge on particle morphology is useful, and in principle, ACSM and DMA data can be combine to retrieve morphology information.

(3) Section 4.1, Lines 347-351: As pointed out by recent studies, amines may contribute significantly to the NPF events (e.g., Zhang et al., Chem Rev., 2015). Several studies have shown that amine compounds in aerosol phase can be hygroscopic, sometimes even at event low RH (e.g., Gomez-Hernandez et al., ES&T, 2016; Chu et al., PCCP, 2015; Qiu and Zhang, ES&T, 2013). Since the reported field measurements took place in a local with heavy industrial activities, it is possible that amine may contribute significantly to the hygroscopicity of the 40-nm particles

(4) Section 4.3: It would make more sense to merge Figures 6&7 as the discussions on the two figures are closely related.

(5) Section 4.4: It seems odd that kappa was not derived from CCN data as described by Petters and Kreidenweis (2007). A side-by-side comparison of kappa values derived from HTDMA, chemical speciation and CCN may be more straightforward. Also, CCN-derived kappa values can also provide basis for comparison with other studies that

may only had CCN results.

---

## Referee Comment (RC2) · Anonymous Referee #1 · 11 Mar 2018

Based on a field campaign conducted in Xingtai in the central North China Plain, the authors discussed about the chemical composition, hygroscopicity and CCN activity of aerosol particles. There have been a number of studies talked about the north part of NCP but very few about the central part. And it was found that aerosol mixing state and hygroscopicity in Xingtai largely differs from that in the north part of NCP. My major concerns are:

1) Xingtai locates at the western boundary of NCP and is heavily affected by the mountain-valley wind (L126). Can the measurement well represent the background aerosol in the central NCP?

2) I did not find a strong connection between the sections of CCN (section 4.3.3 and 4.4) and HTDMA (section 4.1 to 4.3.2). They look like two independent works but each of them a too weak to be an individual study.

Specific comments:

L51: defined as the mixture of solid and liquid particles suspended in air,

L79: I did not see any causal relationship between L72-78 and L79-81

L113: the author need to give more detailed information of the station. Does it locate in urban, sub-urban or rural environment? What about the surroundings, any roads, industrial or residential activities nearby?

L114: how do the authors define the NCP? From the map in Fig. 1 it seems the station locates at the southern boundary of the plain. In L126 the authors also state the station is "heavily affected by the mountain-valley wind".

L138: are then passed – pass

L144: Normally the RH of the HTDMA should be calibrated with ammonium sulfate, especially for high RH measurement. Did the authors calibrate the system during the campaign?

L147: I suggest to use single letter for variables, for example, $f_{g}$ for growth factor.

L154: An Aerosol Chemical. . .

L156: Is the cyclone for all the aerosol instruments or only for the ACSM?

L164: Is there a separate inlet line with PM1 size cut for the aethalometer? I think the authors need to give a clear description of the inlets and sampling line for all the instruments. Now I am a bit confused.

L181: I do not know why the SS needs to be corrected. Normally after applying the calibration parameters to the system, the set SS is the true effective SS in the chamber.

No more correction is needed. Is the SS corrected with the first or second calibration result? Is there any large difference between the two calibrations?

L183: AR defined in the manuscript is determined both on chemical composition and PNSD. I suggest using critical diameter.

L235: I guess here the authors mean the kappa-Sc relationship.

L246: How do the authors define the mean diameter.

L261: Is there undefined species? Maybe the authors can do a simple mass closure between ACSM and SMPS data by assuming a typical aerosol density in that region, to check if there is anything missing in the aerosol mass defined by ACSM. This is important since the authors use this data later to estimate kappa.

L267: How did the authors assume the kappa for HOA, SOA and OOA?

L277: the hydrophobic mode locates at kappa of 0.05 for Dd of 40 nm, and shift towards 0 for large particles. Do the authors have any explanation on this?

L305: the difference decreases with increasing size because most of the larger particles are well aged.

L316: of precursors

L316: is there trace gas (SO2, NOx etc.) measurement during the campaign to support the hypothesis here?

L346: 100 nm is too much for nucleation mode.

L359: From Fig. 6 it seems kappa already starts decrease at 12:00. The secondary aerosol production is also active in the afternoon (Fig. 7b). Why does kappa for larger particles decrease?

L360: in Fig. 6c, kappa of larger particles also reaches ∼0.4 around noon, which is also very close to that of pure AS. But from Fig. 6d we can see there is still a large

fraction of hydrophobic and less-hygroscopic species.

L370: the growth of the newly formed particles is also driven by condensation of sulfate and organics.

L374: why does the PM1 composition differs largely from 40-200 nm particles? 200 nm is in the accumulation mode which is the main contributor of PM1 mass. So I would not expect a large difference between kappa_{chem} and kappa_{gf,200nm}.

L387: Use critical diameter in stead of AR in this section.

L394: the number concentration of fine particles. . .

L400: I do not think one can get this conclusion based on the discussion in this section. The result is in consistent with it but can not directly prove it. Also I think the paper of Dusek et al., (2006) should be cited here. Dusek U, Frank G P, Hildebrandt L, et al. Size matters more than chemistry for cloud-nucleating ability of aerosol particles[J]. Science, 2006, 312(5778):1375.

L407: the critical diameters of aerosol for SS from 0.07% to 0.80% range from about 30 nm to 200 nm. As shown in Fig.6, there is a large difference between kappa_{chem} and kappa_{gf} and the authors claimed that PM1 composition "differ greatly from 40-200 nm particles" (L374). It means one can not use the PM1 composition in the CCN closure. I think a better way might be to compare the kappa obtained from HTDMA and CCN measurements. Although some species may exhibit different kappa in sub- and super-saturation, at least the two measurements are in the same size range.

L434: From this paragraph what I understand is, 0.31 is the average kappa_{chem} and calculated Nccn is not sensitive on the variation of kappa_{chem}. But how can you make sure that 0.31 is a good proxy for the calculation of Nccn in XT? You found some discrepancies between the calculated and measured Nccn and claimed that there are some biases in measured Nccn. So, there is no reference to check which kappa value is appropriate. Normally, water depletion effect can be neglected if Nccn<1e4. Maybe

you can try to do the fit for data points <1e4 and >1e4 separately.

L437: the sensitivity of Nccn on aerosol mixing state is not examined in this section.

---

## Author Comment (AC1) · 28 Mar 2018

**Reply to RC1**

This study reported field measurements results of the chemical speciation, hygroscopicity and CCN properties of ambient particles at a suburban site in the central North China Plain (NCP). The probability density function of the hygroscopicity parameter kappa-PDF was then derived from measurement data and showed only a singular hygrophilic mode which was very different from profiles observed in other regions of the NCP (that were normally bi- or tri-modal). Among the possible factors affecting aerosol hygroscopicity (the mixing state, chemical composition and particle size), particle size was identified as the key factor influencing the particle CCN activation. This study explored the aerosol microphysical properties in a region that was not previously studied and the results can be useful when compared with existing data to understand aerosol aging and its impact on particle microphysics and the climate. The topic is relevant to the scope of the journal of Atmospheric Chemistry and Physics and should be considered for publication.

(1) The tile may be changed to "Characterization of aerosol hygroscopicity, mixing state, and CCN activity at a suburban site in the central North China Plain" to reflect the unique location of this study.

Re: It's a good suggestion, the title has been revised. Very thanks.

(2) Section 4.1, Lines 317-321: While it's been shown that aging of BC will enhance its hygroscopicity and CCN activation, the actual determination of the GF of aged BC could be challenging (see, for example, Torsten et al., Environ. Res. Lett., 2011) as the DMA mobility size change may be marginal. A few chambers studies on soot SOA from anthropogenic VOCs may provide some insights here (Guo et al., ES&T, 2016 and Qiu et al., ES&T, 2012). In general, the knowledge on particle morphology is useful, and in principle, ACSM and DMA data can be combine to retrieve morphology information.

Re: Thanks for your suggestion and the recommended references. We read the recommended papers carefully and indeed found a more comprehensive interpretation for the enhanced hygroscopicity. The related text has been revised as "This suggests that the particles were highly aged and internally mixed at XT during this campaign. Coating of sulfates and secondary organics during the aging process changes the structure of BC and makes it grow, which can significantly enhance the hygroscopicity of particles (e.g., Zhang et al., 2008; Jimenez et al., 2009; Tritscher et al., 2011; Guo et al., 2016).

(3) Section 4.1, Lines 347-351: As pointed out by recent studies, amines may contribute significantly to the NPF events (e.g., Zhang et al., Chem Rev., 2015). Several studies have shown that amine compounds in aerosol phase can be hygroscopic, sometimes even at event low RH (e.g., Gomez-Hernandez et al., ES&T, 2016; Chu et al., PCCP, 2015; Qiu and Zhang, ES&T, 2013). Since the

reported field measurements took place in a local with heavy industrial activities, it is possible that amine may contribute significantly to the hygroscopicity of the 40-nm particles

Re: This is a good point interpreting the hygroscopicity difference of 40 nm particles with other size particles. According to the recommended references, the text has been revised as "40 nm particles were always more hygroscopic than 80 nm particles at XT, especially in the daytime, which was also different from other sites. This is likely because the coating effect of sulfates and secondary organics is more significant on smaller particles (Tritscher et al., 2011; Guo et al., 2016). Furthermore, since the field measurements took place in a local with heavy industrial activities, it is possible that amine contributes significantly to the hygroscopicity of 40 nm particles. Several studies have shown that amine compounds in aerosol phase can be hygroscopic, sometimes at even low RH (Qiu et al., 2012; Chu et al., 2015; Gomez-Hernandez et al., 2016).".

(4) Section 4.3: It would make more sense to merge Figures 6&7 as the discussions on the two figures are closely related.

Re: It's a good suggestion, we have merged them, as:

[Figure]

Indeed, the merging makes more sense. It is more clear to note that the increase of hygroscopicity parameter ($\kappa_{gf}$) in the morning was synchronous with the particle number concentration ($N_{15\text{-}685\ nm}$), but was not with the PBL height, further suggesting the impact of photochemical reactions on aerosol hygroscopicity.

(5) Section 4.4: It seems odd that kappa was not derived from CCN data as described by Petters and Kreidenweis (2007). A side-by-side comparison of kappa values derived from HTDMA, chemical speciation and CCN may be more straightforward. Also, CCNderived kappa values can also provide basis for comparison with other studies that may only had CCN results.

Re: Yes, we will obtain more information about aerosol hygroscopicity if kappa from CCN data can be derived. However, it needs to connect DMA and CCNc to measure the size-CCN number concentration. Unfortunately, we only measured the bulk CCN number concentration in the campaign. We will do the work in the future campaigns.

**References:**

Chu Y., Sauerwein M. and Chan C.K.: Hygroscopic and phase transition properties of alkyl aminium sulfates at low relative humidities, Phys Chem Chem Phys, 17, 19789-19796, https://doi.org/10.1039/c5cp02404h, 2015.

Gomez-Hernandez M., McKeown M., Secrest J., Marrero-Ortiz W., Lavi A., Rudich Y., Collins D.R. and Zhang R.: Hygroscopic Characteristics of Alkylaminium Carboxylate Aerosols, Environ Sci Technol, 50, 2292-2300, https://dx.doi.org/10.1021/acs.est.5b04691, 2016.

Guo S., Hu M., Lin Y., Gomez-Hernandez M., Zamora M.L., Peng J., Collins D.R. and Zhang R.: OH-Initiated Oxidation of m-Xylene on Black Carbon Aging, Environ Sci Technol, 50, 8605-8612, https://dx.doi.org/10.1021/acs.est.6b01272, 2016.

Qiu C. and Zhang R.: Physiochemical Properties of Alkylaminium Sulfates: Hygroscopicity, Thermostability, and Density, Environ Sci Technol, 46, 4474-4480, https://dx.doi.org/10.1021/es3004377, 2012.

Tritscher T., Juranyi Z., Martin M., Chirico R., Gysel M., Heringa M.F., DeCarlo P.F., Sierau B., Prevot A.S.H., Weingartner E. and Baltensperger U.: Changes of hygroscopicity and morphology during ageing of diesel soot, Environ. Res. Lett., 6, https://dx.doi.org/10.1088/1748-9326/6/3/034026, 2011.

---

## Author Comment (AC2) · 28 Mar 2018

**Reply to RC2**

Based on a field campaign conducted in Xingtai in the central North China Plain, the authors discussed about the chemical composition, hygroscopicity and CCN activity of aerosol particles. There have been a number of studies talked about the north part of NCP but very few about the central part. And it was found that aerosol mixing state and hygroscopicity in Xingtai largely differs from that in the north part of NCP. My major concerns are:

1) Xingtai locates at the western boundary of NCP and is heavily affected by the mountain-valley wind (L126). Can the measurement well represent the background aerosol in the central NCP?

Re: The influence of mountain-valley wind is regional. Figure 1 in this reply shows pollution in the NCP is rather non-uniform, the central region is more serious than that in northern and southern regions. Our sampling site (Xingtai) is located in one of the pollution centers and thus represents the condition near an emission source region. The average $PM_{2.5}$ mass concentration at this station was 45.2 μg m$^{-3}$ in this campaign, which was only 15% lower than that (53.3 μg m$^{-3}$) measured at the urban site in Xingtai. In addition, OOA and sulfate were the most important organic and inorganic species respectively. Figure 2 in this reply shows that the diurnal cycle of OOA was flat and the diurnal cycle of sulfate was also smoother compared with nitrate, also reflecting the regional characteristics of the main pollutants. All these suggest that the influence of mountain-valley wind is limited and this site is a good representative in this region. Detailed discussion about gas precursors and aerosol chemical species in this region can be found in Zhang et al., (2018).

[Figure]

Figure 1. The distribution of mean $SO_2$ concentrations of May from 2012 to 2016. The map of NCP can refer to Fig. 4 in this reply.

[Figure]

Figure 2. Diurnal cycles of chemical species of $PM_1$ and OA factors during entire study, PE (polluted events) and CP (clean periods). The figure is from Zhang et al., (2018).

2) I did not find a strong connection between the sections of CCN (section 4.3.3 and 4.4) and HTDMA (section 4.1 to 4.3.2). They look like two independent works but each of them a too weak to be an individual study.

Re: Aerosol mixing state is one of three factors influencing aerosol activation ability. In this campaign, we only measured bulk CCN data rather than size-resolved CCN data. The bulk CCN data cannot provide the information about aerosol mixing state but HTDMA data can. HTDMA data in this campaign suggests the aerosol is highly internal-mixed and aged, which is the base to do CCN closure studies in this paper. In addition, the HTDMA data are very useful for the analysis of the activation rate diurnal variations. These are important links between the sections of HTDMA and CCN.

Specific comments:

L51: defined as the mixture of solid and liquid particles suspended in air,

Re: Good suggestion, we have corrected.

L79: I did not see any causal relationship between L72-78 and L79-81

Re: L79-81 had been corrected as "aerosol hygroscopicity and CCN activity are very different in different regions due to different chemical compositions".

L113: the author need to give more detailed information of the station. Does it locate in urban, sub-urban or rural environment? What about the surroundings, any roads, industrial or residential activities nearby?

Re: This suburban site is situated ~17 km northwest of Xingtai urban area in southern Hebei Province. A provincial road is ~400 m southeast of the sample site, a school adjoins the east, and a town is ~600 m in the northwest (Fig. 3 in this reply). There are many industrial manufacturers in this region including coal-based power plants, steel and iron works, glassworks, and cement mills. We have added more environment information about the sample site in the manuscript.

[Figure]

Figure **3**. The surroundings at the sample site.

L114: how do the authors define the NCP? From the map in Fig. 1 it seems the station locates at the southern boundary of the plain. In L126 the authors also state the station is "heavily affected by the mountain-valley wind".
Re: NCP is one of the largest plains in China. It covers an area of about 0.3 million square kilometers. The Fig.1 in the manuscript only shows a part of NCP, a full image can refer to Fig. 4 in this reply. This figure shows the observation site is in the central NCP although it is not far away from the Taihang Mountains. This station is still a good representative site in this region although it is affected by the mountain-valley wind (c.f. response to the Question 1).

[Figure]

Figure **4**. Map showing the observation site and the scale of North China Plain (NCP). The original figure is from Wang et al., (2018).

L138: are then passed – pass

Re: Yes, we have corrected it.

L144: Normally the RH of the HTDMA should be calibrated with ammonium sulfate, especially for high RH measurement. Did the authors calibrate the system during the campaign?

Re: Yes, the RH calibration is important for HTDMA running. Figure 5 in this reply shows the calibration result with ammonium sulfate during our campaign. It shows the deliquescence point of ammonium sulfate is 78±1 %, this is consistent with previous studies (Badger et al., 2006; Tan et al., 2013).

[Figure]

Figure **5**. Humidogram of ammonium sulfate for 150-nm particles measured with the HTDMA.

L147: I suggest to use single letter for variables, for example, f_{g} for growth factor.

Re: It's a good suggestion, but we tried to follow the conventional acronyms such as "GF" or "*gf*" that has been used in most previous papers. "$f_g$" is a good suggestion but it's easily confused with the expression of the aerosol scattering enhancement factor ($f_{RH}$). Therefore, we'd prefer to continue the use of "GF" in this manuscript.

L154: An Aerosol Chemical. . .

Re: Corrected per the comment.

L156: Is the cyclone for all the aerosol instruments or only for the ACSM?

Re: ACSM and AE33 had their separate cyclones. SMPS had an impactor in the inlet, which can also remove the most particles larger than its measurement range. HTDMA had no cyclone which has a minor effect on its measurement. CCNc had not also a cyclone because it needed to measure the bulk CCN number concentration in this campaign.

L164: Is there a separate inlet line with PM1 size cut for the aethalometer? I think the authors need to give a clear description of the inlets and sampling line for all the instruments. Now I am a bit confused.

Re: Yes, aethalometer had a separate inlet headed by a $PM_1$ for filtering particle size great than 1μm. CCNc and HTDMA shared the same inlet. During this campaign, all sampling instruments were placed in two containers at ground level and two air conditioners were used to maintain the temperature at 20–25 ºC inside containers. All stainless tube inlets were 1.5 m above the top of containers.

L181: I do not know why the SS needs to be corrected. Normally after applying the calibration parameters to the system, the set SS is the true effective SS in the chamber. No more correction is needed. Is the SS corrected with the first or second calibration result? Is there any large difference between the two calibrations?

Re: Actually, the SS is influenced by flow rates and the temperature gradient in the cloud column. Therefore, the flow and temperature calibrations are also needed, which had been conducted before this campaign and the corresponding parameters were applied in the system. However, the SS maybe changed if the CCNc runs unsteadily, so we did two SS campaigns before and after the campaign instead of using the SS parameters from a single calibration. Figure 6 in this reply shows the results of the SS calibrations, the calibration method is the same as Rose et al., (2008). The results show CCNc run well and steadily during this campaign. The calibrated SS used in this paper was from the mean SS of two calibration results.

[Figure]

Figure **6**. The results of SS calibration experiments with ammonium sulfate: CCN efficiency spectra measured at 5 different temperature gradient ($\Delta T$). SS_cal1 and SS_cal2 are the calibration results before and after the campaign respectively.

L183: AR defined in the manuscript is determined both on chemical composition and PNSD. I suggest using critical diameter.

Re: The critical diameter ($D_{0,\text{crit}}$) used in this study was derived from the hygroscopicity parameter ($\kappa_{\text{chem}}$), not measured directly. AR is used for a preliminary analysis of its relationships with nucleation events and PBL height. In the further campaigns, we will conduct the size-resolved CCN measurements through connecting DMA and CCNc. The critical diameter will be retrieved from size-CCN data, which can study the relationships in detail.

L235: I guess here the authors mean the kappa-Sc relationship.

Re: The relationships of $\kappa$, $S_c$ ($S_c = s_c - 1$) and $D_d$ is shown in Fig.1 in Petters et al. (2007). Actually, the $D_d$ is the critical diameter corresponding to the critical supersaturation when $\kappa$ is known. Therefore, here we first establish the $s_c$-$D_d$ relationship, not $\kappa$-$s_c$ relationship.

L246: How do the authors define the mean diameter.

Re: Here the mean diameter is the geometric mean diameter ($D_m$). It is defined as:

$$D_m = \frac{\int_{15nm}^{685nm} n(\log D_p) D_p \, d\log D_p}{\int_{15nm}^{685nm} n(\log D_p) \, d\log D_p}$$

We found a mistake in its calculation previously, now it has been corrected in the revised manuscript for which we are very grateful to the reviewer.

L261: Is there undefined species? Maybe the authors can do a simple mass closure between ACSM and SMPS data by assuming a typical aerosol density in that region, to check if there is anything missing in the aerosol mass defined by ACSM. This is important since the authors use this data later to estimate kappa.

Re: The mass closure between ACSM and SMPS is hard to realize because aerosol density and morphology are unknown during this campaign. Even so, we try to do a simple mass closure according to the suggestion of the reviewer. Here we assume aerosol particles are spherical and the density is 1.6 g/cm$^3$. The closure result (Fig. 8 in this reply) shows the two masses are related dependent on time. For most of the time, their correlation is good although the SMPS mass was always lower than that of ACSM+BC. The biases are likely caused by different measurement size range, and the variations of aerosol density and morphology.

[Figure]

Figure **7**. Comparison of the mass concentrations of PM$_1$ (=NR-PM$_1$+BC) with those derived from SMPS measurements (D$_p$ = 15-685 nm).

L267: How did the authors assume the kappa for HOA, SOA and OOA?

Re: The gravimetric densities ($\rho$) and hygroscopicity parameters ($\kappa$) of all species used in this study for CCN closure (note: POA = HOA+COA, SOA = OOA).

| Species | NH$_4$NO$_3$ | (NH$_4$)$_2$SO$_4$ | NH$_4$HSO$_4$ | H$_2$SO$_4$ | POA | SOA | BC |
|---|---|---|---|---|---|---|---|
| $\rho$ (kg m$^{-3}$) | 1720 | 1769 | 1780 | 1830 | 1000 | 1400 | 1700 |
| K | 0.67 | 0.61 | 0.61 | 0.9 | 0 | 0.1 | 0 |

L277: the hydrophobic mode locates at kappa of 0.05 for Dd of 40 nm, and shift towards 0 for large particles. Do the authors have any explanation on this?

Re: The direct comparison of the κ-PDF is depicted in Fig. 9 in this reply. There is an obvious difference of the hydrophobic mode as the reviewer found. This is related to the different chemical compositions. The hydrophobic mode of 40-nm particles is mainly caused by organics, while that of the larger particles is mainly caused by BC. BC is fully hydrophobic but some organics has a limited water uptake ability. This can be verified from previous studies, Wu et al., (2017) reported the fresh emitted BC are mainly in the accumulation mode.

[Figure]

Figure 8. Comparison of κ-PDF of different size particles.

L305: the difference decreases with increasing size because most of the larger particles are well aged.

Re: Good point. We have added the corresponding expression.

L316: of precursors

Re: We have corrected.

L316: is there trace gas (SO2, NOx etc.) measurement during the campaign to support the hypothesis here?

Re: Yes. Figure 10 in this reply shows the diurnal variations of trace gases and meteorological variables ($T$ and RH) in this campaign. Affected by the mountain-valley wind, prevailing winds shifted from the northwest to the southeast in the early morning. There are more industrial emissions to the southeast of the measurement site than to the northwest. Therefore, the CO, $SO_2$ concentrations increased sharply in the morning after the wind shift. The increased CO suggests the increase of VOCs because of their similar sources. The $O_3$ concentration increased gradually after sunrise during the day, reflecting the enhancement of atmospheric oxidation capacity. The ample supply of effluent $SO_2$ and VOCs precursors and the strong atmospheric oxidation capacity under high RH and low T conditions made plenty of sulfate and SOA produce (Wang et al., 2016; Wang et al., 2017). This is the reason in the frequent occurrence of NPF events and the enhancement of aerosol hygroscopicity during the daytime at XT.

[Figure]

Figure **9**. Diurnal variations in trace gases (CO, NOx/NO, SO2 and O3) and meteorological variables (*T* and RH).

L346: 100 nm is too much for nucleation mode.
Re: It has been corrected as "small Aitken mode particles (< 50 nm)".
L359: From Fig. 6 it seems kappa already starts decrease at 12:00. The secondary aerosol production is also active in the afternoon (Fig. 7b). Why does kappa for larger particles decrease?
Re: This is related with the local emissions, especially the emitted organics. As is shown in the diurnal variations (Fig.2 in this reply), the primary organics (COA and HOA) increase sharply near noon. This is the reason why κ has a short-time decrease at near 12:00.
L360: in Fig. 6c, kappa of larger particles also reaches _0.4 around noon, which is also very close to that of pure AS. But from Fig. 6d we can see there is still a large fraction of hydrophobic and less-hygroscopic species.
Re: As discussed in section 4.2, aerosol particles in this site were highly internal-mixed and aged. κ-PDF for all size particles showed only one hydrophilic mode during daytime although there was still a large fraction of hydrophobic and less-hygroscopic species. This is because the condensation of sulfuric acid on organics or BC can greatly enhance their hygroscopicity (Peng et al., 2016; Zhang et al., 2017).
L370: the growth of the newly formed particles is also driven by condensation of sulfate and organics.
Re: Yes, we have corrected.
L374: why does the PM1 composition differs largely from 40-200 nm particles? 200 nm is in the accumulation mode which is the main contributor of PM1 mass. So I

would not expect a large difference between kappa_{chem} and kappa_{gf,200nm}.

Re: The reviewer is right. We have deleted the expression "the bulk chemical compositions of $PM_1$ and of 40–200 nm particles differ greatly" in the manuscript. The difference was mainly induced by the simple ZSR mixing rule. This feature was stronger at noon when atmospheric oxidation and the aging process were more rapid. We have studied this phenomenon deeply in another paper (Zhang et al., 2017).

L387: Use critical diameter in stead of AR in this section.

Re: As mentioned above, we did not take size-resolved CCN measurement in this campaign, so critical diameter cannot be got directly. We will do the work in the future campaigns.

L394: the number concentration of fine particles. . .

Re: We have corrected.

L400: I do not think one can get this conclusion based on the discussion in this section. The result is in consistent with it but can not directly prove it. Also I think the paper of Dusek et al., (2006) should be cited here. Dusek U, Frank G P, Hildebrandt L, et al. Size matters more than chemistry for cloud-nucleating ability of aerosol particles[J]. Science, 2006, 312(5778):1375.

Re: Figure 11 in this reply depicts the sensibility tests of aerosol size distribution and chemical composition to the CCN number concentration estimation, which are similar with that of Dusek et al., (2006). The figure shows the closure correlation using mean NASD (normalized aerosol size distribution) is weaker than that using mean $\kappa_{chem}$, reflecting the more important effect of particle size for aerosol CCN activity than chemical composition. We have added the figure in the supplement and cited this paper in the revised manuscript.

[Figure]

Figure **10**. Correlation between the measured and estimated CCN number concentration. The later was from the mean NASD (normalized aerosol size distribution) but variable composition ($\kappa_{chem}$) or using the mean $\kappa_{chem}$ but variable NASD.

L407: the critical diameters of aerosol for SS from 0.07% to 0.80% range from about 30 nm to 200 nm. As shown in Fig.6, there is a large difference between kappa_{chem} and kappa_{gf} and the authors claimed that PM1 composition "differ greatly from 40-200 nm particles" (L374). It means one can not use the PM1 composition in the CCN closure. I think a better way might be to compare the kappa obtained from HTDMA and CCN measurements. Although some species may exhibit different kappa in sub- and super-saturation, at least the two measurements are in the same size range.

Re: The related expression "$PM_1$ composition differ greatly from 40-200 nm particles" is not appropriate, which has been deleted in this manuscript. The $\kappa$ closure from HTDMA and CCN data is useful to analyze the aerosol hygroscopicity in detail. However, it cannot be realized in this campaign because no size-resolved CCN data. It will be done in the future campaigns.

L434: From this paragraph what I understand is, 0.31 is the average kappa_{chem} and calculated Nccn is not sensitive on the variation of kappa_{chem}. But how can you make sure that 0.31 is a good proxy for the calculation of Nccn in XT? You found some discrepancies between the calculated and measured Nccn and claimed that there are some biases in measured Nccn. So, there is no reference to check which kappa value is appropriate. Normally, water depletion effect can be neglected if Nccn<1e4. Maybe you can try to do the fit for data points <1e4 and >1e4 separately.

Re: Figure 8 in the manuscript shows there is a limited difference for the CCN closure using variable or mean $\kappa_{chem}$, so $N_{CCN}$ is not sensitive on the variation of $\kappa_{chem}$. 0.31 is only a reference value for people who need to calculate the CCN concentration in this region in their models. As the reviewer says, the CCN closure can be categorized according to $N_{CCN}$. Actually, the measured $N_{CCN}$ is biased when $N_{CCN} > 5500$ cm$^{-3}$ (Fig. 12 in this reply), this is coincident with the report from DMT-CCNC manual. Figure 12 in this reply also shows the CCN closure is very good when $N_{CCN} < 5500$ cm$^{-3}$, reflecting the validation of the closure method used in this study.

[Figure]

Figure 11. Estimated versus measured CCN number concentrations at SS = 0.75 %. The $N_{CCN}$ is estimated based on κ-Köhler theory, using the real-time $\kappa_{chem}$. Here, the critical value of $N_{CCN}$ = 5500 cm$^{-3}$ is used to separate the points into two groups. A separate linear regression analysis is done on each group. The slopes, correlation coefficients ($R^2$), and relative deviations (RD) are shown in the figure.

L437: the sensitivity of Nccn on aerosol mixing state is not examined in this section.
Re: According to the HTDMA measurement results, aerosols in this region are highly internal-mixed and aged. Therefore, we directly assume aerosols are internally mixed when calculating the CCN number concentration as description in section 3.2.
Another work of our group have suggested the mixing state had a minor effect for the CCN estimation in Beijing (Ren et al., 2017).

**References:**
Badger C.L., George I., Griffiths P.T., Braban C.F., Cox R.A. and Abbatt J.P.D.: Phase transitions and hygroscopic growth of aerosol particles containing humic acid and mixtures of humic acid and ammonium sulphate, Atmos Chem Phys, 6, 755-768, 10.5194/acp-6-755-2006, 2006.

Dusek U., Frank G.P., Hildebrandt L., Curtius J., Schneider J., Walter S., Chand D., Drewnick F., Hings S. and Jung D.: Size matters more than chemistry for cloud-nucleating ability of aerosol particles., Science, 312, 1375-8, 2006.

Peng J., Hu M., Guo S., Du Z., Zheng J., Shang D., Zamora M.L., Zeng L., Shao M. and Wu Y.: Markedly enhanced absorption and direct radiative forcing of black carbon under polluted urban environments, Proceedings of the National Academy of Sciences, 113, 4266-4271, 2016.

Petters M.D. and Kreidenweis S.M.: A single parameter representation of hygroscopic growth and cloud condensation nucleus activity, Atmos Chem Phys, 7, 1961-1971, 2007.

Ren J., Zhang F., Wang Y., Fan X., Jin X., Xu W., Sun Y., Cribb M. and Li Z.: Using different assumptions of aerosol mixing state and chemical composition to predict CCN concentrations based

on filed measurement in Beijing, Atmos. Chem. Phys. Discuss., 2017, 1-40, 10.5194/acp-2017-806, 2017.

Rose D., Gunthe S.S., Mikhailov E., Frank G.P., Dusek U., Andreae M.O. and Pöschl U.: Calibration and measurement uncertainties of a continuous-flow cloud condensation nuclei counter (DMT-CCNC): CCN activation of ammonium sulfate and sodium chloride aerosol particles in theory and experiment, Atmos Chem Phys, 8, 1153-1179, 2008.

Tan H., Xu H., Wan Q., Li F., Deng X., Chan P.W., Xia D. and Yin Y.: Design and application of an unattended multifunctional H-TDMA system, J. Atmos Ocean Tech, 30, 1136-1148, 2013.

Wang G., Zhang R., Gomez M.E., Yang L., Zamora M.L., Hu M., Lin Y., Peng J., Guo S. and Meng J.: Persistent sulfate formation from London Fog to Chinese haze, Proceedings of the National Academy of Sciences, 113, 13630-13635, 2016.

Wang Y., Zhao C., Dong Z., Li Z., Hu S., Chen T., Tao F. and Wang Y.: Improved retrieval of cloud base heights from ceilometer using a non-standard instrument method, Atmos Res, 202, 148-155, https://doi.org/10.1016/j.atmosres.2017.11.021, 2018.

Wang Z., Wu Z., Yue D., Shang D., Guo S., Sun J., Ding A., Wang L., Jiang J. and Guo H.: New particle formation in China: Current knowledge and further directions, Sci Total Environ, 577, 258-266, 2017.

Wu Y., Wang X., Tao J., Huang R., Tian P., Cao J., Zhang L., Ho K.F., Han Z. and Zhang R.: Size distribution and source of black carbon aerosol in urban Beijing during winter haze episodes, Atmos Chem Phys, 17, 7965-7975, 10.5194/acp-17-7965-2017, 2017.

Zhang F., Wang Y., Peng J., Ren J., Collins D., Zhang R., Sun Y., Yang X. and Li Z.: Uncertainty in predicting CCN activity of aged and primary aerosols, Journal of Geophysical Research: Atmospheres, 122, 2017.

Zhang Y., Du W., Wang Y., Wang Q., Wang H., Zheng H., Zhang F., Shi H., Bian Y., Han Y., Fu P., Canonaco F., Preˆvoˆt A.S.H., Zhu T., Wang P., Li Z. and Sun Y.: Aerosol chemistry and particle growth events at an urban downwind site in the North China Plain, Atmos. Chem. Phys. Discuss., 2018, 1-29, 10.5194/acp-2017-889, 2018.

---

## Author Response (AR2)

Thanks two reviewers for further reviewing our manuscript. We done some changes in this paper according to their comments. We also improved English language in the new version of this paper.

**Reply to Report #1.**

It appears that the authors has addressed most of my concerns during the initial manuscript review. While the study did not measure size-resolved CCN data which makes it difficult to compare k values derived from side-by-side HTDMA and CCN measurements, it is still the first report about the aerosol CCN properties in a region that was not previously studied. As a result, I recommend the revised manuscript published in the journal Atmospheric Chemistry and Physics.
The other reviewer provided highly detailed comments to further improve the manuscript and the authors have addressed most of them. However, the response table for comment on L267 should be incorporated into the supplementary documents.
Re: Good suggestion, we have added the corresponding table in the supplementary documents.

**Reply to Report #2.**

For the response to general comment 2: I did not say that there is no connection between HTDMA and CCN measurements. What I suggested is to make a smoother transition from the HTDMA part (4.1 to 4.3.2) to CCN part (4.3.3 and 4.4).
Re: Good suggestion. We done some improvement, such as adding the sentence "It is reasonable to assume that aerosols are internally mixed when estimating $N_{CCN}$ because H-TDMA data showed that this was the case at XT." at beginning of section 4.4.

For the response to specific comments L156 and L164: I suggest the authors also add this information to the main text to help audience better understand your measurements.
Re: Thanks for the suggestion. We have added the corresponding sentences in the manuscript.

For the response to specific comment L181: from the information provided in the main text and this response, what I understand is: Calibration of flow and SS was "conducted before this campaign and the corresponding parameters were applied in the system". Then, "Five SS levels, i.e., 0.07, 0.1, 0.2, 0.4, and 0.8 %, were set in the CCNC". Another SS calibration was done after the campaign and "The calibrated SS used in this paper was from the mean SS of two calibration results". "The corrected SS levels were 0.11, 0.13, 0.22, 0.40, and 0.75 %, respectively". It means that with the five deltT (calculated internally in CCN according to the calibration parameters from first calibration), the actual SS changed from the original values (0.07, 0.1, 0.2, 0.4, and 0.8 %)

before the campaign to 0.15%, 0.16%, 0.24%, 0.4% and 0.7% at the end of the campaign. But this
is not what I saw in the calibration curves shown in the response.
This is also why I suggested another "major revisions". I think the authors should clarify this
before the manuscript can be considered for final publication.
Re: We are sorry that the response to specific comment L181 confused the reviewer. Actually, the
flow and temperature sensors were calibrated before this campaign and their corresponding
parameters were used in the system. The SS calibration is different from these calibrations, as SS
is related with the temperature gradient ($\Delta T$) in the cloud chamber, not a certain temperature. We
didn't change the corresponding parameters to SS although we calibrated it before the campaign.
Figure 1 in this reply shows the results of two SS calibrations, suggesting a very limited change of
the relationship between SS and $\Delta T$ before and after the campaign. This verifies that our CCN
counter performed steadily during this campaign.

[Figure]

Figure 1. The results of SS calibration experiments with ammonium sulfate: CCN efficiency
spectra measured at 5 different temperature gradient ($\Delta T$). SS_before and SS_after are the
calibration results before and after the campaign respectively.
For the response to specific comment L183 and L387: with measured PNSD and CCN total
number concentration, critical diameter can be calculated as the diameter above which the
integration of PNSD equals to the CCN number concentration. This treatment has been used in
several studies (e.g. Deng et al., 2013). The advantage is it excludes the influence of the variation
of PNSD in the inferred CCN activities, compared with AR.
Re: The reviewer suggests an alternative method to calculate the critical diameter ($D_c$), so that the
corresponding hygroscopicity parameter ($\kappa_{CCN}$) can be calculated. However, the shortage of the
method lies in that the $D_c$ won't be accurate if the CCN number concentrations ($N_{CCN}$) have biases.
A minor change of $D_c$ will result in a significant change of $\kappa_{CCN}$ because of the strong sensibility
($\kappa_{CCN} \sim D_c^{-3}$). Figure 2 in this reply shows the $\kappa$ values from SMPS-CCNc data using the
recommended method and HTDMA data in this campaign. It's obvious that $\kappa_{CCN}$ is larger than
$\kappa_{HTDMA}$, likely due to the CCNc measurement uncertainties as stated in the manscript. Lower
measured $N_{CCN}$ than its actual value in this polluted environment leads to the overestimation of $D_c$, then will make a underestimation of $\kappa_{CCN}$. This influence is stronger for higher SS (lower $D_c$) due
to higher biases in $N_{CCN}$, which is also reflected in Fig. 2. In a word, this method maybe not
suitable in our data.
Note that our main objective in in L387 is to infer the influence of PBL on the aerosol
activation ability. The influence includes the impact of PBL on PNSD, so we think it's appropriate
to use AR in this paper.

[Figure]

Figure 2. The comparison of hygroscopicity parameter ($\kappa$) retrieved from SMPS-CCNc or
HTDMA data.
For the response to specific comment L407: I think deleting the sentence will not change the
reality that "PM1 composition differ greatly from 40-200 nm particles" as reflected in fig 6.
Re: Yes, it's a good suggestion, we have deleted the corresponding sentence.
For the response to specific comment L434: What I want to point out is, here you can not really
prove that 0.31 is a good proxy for the calculation of Nccn. Because as shown in fig. 6, PM1
composition differ greatly from 40-200 nm particles. Probably a kappa of 0.25 or 0.35 can bring
similar results just because Nccn is not sensitive on kappa.
Re: Agree, but here we only want to provide a reference value for people who need to calculate the
CCN concentration in this region in their models. We have corrected the sentence as "$\kappa = 0.31$
which is a good reference value to model the CCN number concentration in this region".
For the response to specific comment L437: I was not against this statement. I fully agree with it.
What I wanted to say is you should not put it in your conclusion since you did not prove it in this
section.
Re: Agree. The corresponding sentence about mixing state in the conclusion has been deleted.

[revised manuscript text omitted]

---

## Author Response (AR3)

The authors have addressed most of my comments. However, I still have some questions about the calibration curve (specific comment L181).

In the response the authors tried to explain the calibration procedure but did not answer my questions directly. I've been a user of CCN counter since 8 years and I know how to calibrate it, but I am still a bit confused.

In the last response, the authors wrote "We didn't change the corresponding parameters to SS although we calibrated it before the campaign". I am wondering why the authors did not apply the new calibration curve. It means an older curve was used to convert the set SS to deltTs, and these deltTs were used to calculate the corrected SS, right? Can you add also the older calibration curve into the figure?

Re:We are sorry that our response confused the reviewer. Actually, our meaning is that we did not change the set parameters (Temp Gradient Slope and Temp Gradient Intercept) in the "CCN Calibration Editor.vi" software before the campaign. New linear curves between effective supersaturation ($SS_{eff}$) and $\Delta T$ were established before and after the campaign (Fig. 1) and their mean curve was used in this study. There is a one-to-one correspondence between the set SS and the set $\Delta T$. We calculated the corrected SS utilizing the new curve and the set $\Delta T$. Therefore, the corrected SS (0.11, 0.13, 0.22, 0.40, and 0.75%) were different with the set SS (0.07, 0.1, 0.2, 0.4, and 0.8%). The corrected SS were used in this study. Same calibrating procedures were reported by Deng et al. (2011) and Zhang et al. (2014).

We are sorry we cannot show the older calibration curve because we can't find the old data. We never used the older curve in this study. There was a small difference for the relationship between $SS_{eff}$ and $\Delta T$ before and after this campaign (Fig. 1), suggesting the new curve was reliable and the CCNc worked well during this campaign.

[Figure]

Figure 1. The results of SS calibration experiments with ammonium sulfate: CCN efficiency spectra measured at 5 different temperature gradient ($\Delta T$). SS_before and SS_after are the calibration results before and after the campaign respectively.

By the way, I am also very curious about how the author calibrated the temperature sensors in CCN counter. I think this is not part of the standard calibration procedure which an user is supposed to do. Can you explain how you did it?

Re:The user cannot do the temperature sensor calibration because it needs some special instruments. A technical company help us do the calibration. The sensor performance can be adjusted through regulating resistance. The engineer first puts CCNc in a thermostatic container, and then adjusts these resistors to make the sensor temperatures consistent with the standard temperature inside the container.

**References**

[revised manuscript text omitted]